

# Integrable hierarchies, Hurwitz numbers and a branch point field in critical topologically massive gravity

Yannick Mvondo-She

Department of Physics, University of Pretoria, Private Bag X20, Hatfield 0028, South Africa

vondosh7@gmail.com

## Abstract

We discuss integrable aspects of the logarithmic contribution of the partition function of cosmological critical topologically massive gravity. On one hand, written in terms of Bell polynomials which describe the statistics of set partitions, the partition function of the logarithmic fields is a generating function of the potential Burgers hierarchy. On the other hand, the polynomial variables are solutions of the Kadomtsev-Petviashvili equation, and the partition function is a KP $\tau$ function, making more precise the solitonic nature of the logarithmic fields being counted. We show that the partition function is a generating function of Hurwitz numbers, and derive its expression. The fact that the partition function is the generating function of branched coverings gives insight on the orbifold target space. We show that the logarithmic field $\psi_{\mu\nu}^{new}$ can be regarded as a branch point field associated to the branch point $\mu l = 1$.

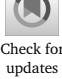

# 1 Introduction

Gravity in three dimensions has for some time now been an interesting model to test theories of -classical and quantum- gravity, and a consistent non-trivial theory would bring the prospect of clarifying many intricate aspects of gravity. A fundamental breakthrough was made in the study of the asymptotics, revealing the emergence of a Virasoro algebra at the boundary [1]. One can thus expect a dual 2d CFT description, and this discovery can be thought of as a precursor of the AdS/CFT correspondence. However, pure Einstein gravity in three dimensions is locally trivial at the classical level and does not exhibit propagating degrees of freedom. Hence there was a need to modify it.

One way of modifying pure Einstein gravity is by introducing a negative cosmological constant. Although the resulting theory still has no propagating degrees of freedom, it has black hole solutions [2]. Another possible modification is to add a gravitational Chern-Simons term. In that case the theory is called topologically massive gravity (TMG), and contains a propagating degree of freedom, the massive graviton [3,4]. When both cosmological and Chern-Simons terms are included in a theory, it yields cosmological topologically massive gravity (CTMG). Such a theory features both gravitons and black holes.

Following Witten's proposal in 2007 to find a CFT dual to Einstein gravity [5], the graviton 1-loop partition function was calculated in [6]. However, discrepancies were found in the results. In particular, the left- and right-contributions did not factorize, therefore clashing with the proposal of [5].

Soon after, a non-trivial slightly modified version of Witten's construct was proposed by Li, Song and Strominger [7]. Their theory, in which Einstein gravity was replaced by *chiral* gravity can be viewed as a special case of topologically massive gravity [3, 4], at a specific tuning of the couplings, and is asymptotically defined with AdS$_3$ boundary conditions, in the spirit of Fefferman-Graham-Brown-Henneaux [1,8,9]. A particular feature of the theory was that one of the two central charges vanishes, whilst the other one can have a non-zero value. This gave an indication that the partition function could factorize.

Shortly after the proposal of [7], Grumiller *et al.* noticed that relaxing the Brown-Henneaux boundary conditions allowed for the presence of a massive mode that forms a Jordan cell with the massless graviton, leading to a degeneracy at the critical point [10]. In addition, it was observed that the presence of the massive mode spoils the chirality of the theory, as well as its unitarity. Based on these results, the dual CFT of critical cosmological topologically massive gravity (cTMG) was conjectured to be logarithmic, and the massive mode was called the *logarithmic partner* of the graviton. Indeed, Jordan cell structures are a noticeable feature of logarithmic CFTs, that are non unitary theories (see [11] as well as the very nice introductory notes [12] and [13]). The correspondence distinguishes itself on the conjectured dual LCFT side by a left-moving energy-momentum tensor $T$ that has a logarithmic partner state $t$ with identical conformal weight, forming the following Jordan cell

$$L_0 \begin{pmatrix} |T\rangle \\ |t\rangle \end{pmatrix} = \begin{pmatrix} 2 & 0 \\ 1 & 2 \end{pmatrix} \begin{pmatrix} |T\rangle \\ |t\rangle \end{pmatrix}. \tag{1}$$

A major achievement towards the formulation of the correspondence was the calculation of correlation functions in TMG [14,15], which confirmed the existence of logarithmic correlators of the type $\langle T(x)t(y)\rangle = b_L/(x-y)^4$ that arise in LCFT, with $b_L$ commonly referred as the logarithmic anomaly. Subsequently, the 1-loop graviton partition function of cTMG on the thermal AdS$_3$ background was calculated in [16], resulting in the following expression

$$Z_{\text{cTMG}}(q,\bar{q}) = \prod_{n=2}^{\infty} \frac{1}{|1-q^n|^2} \prod_{m=2}^{\infty} \prod_{\bar{m}=0}^{\infty} \frac{1}{1-q^m\bar{q}^{\bar{m}}}, \qquad \text{with } q = e^{2i\pi\tau}, \bar{q} = e^{-2i\pi\bar{\tau}}, \tag{2}$$

where the first product can be identified as the three-dimensional gravity partition function $Z_{0,1}$ in [6], and is therefore not modular invariant. The double product is the partition function of the logarithmic single and multi particle logarithmic states, and will be the central object of this work.

The corresponding expression of the partition function on the CFT side was derived in [16] and given the form

$$Z_{\text{LCFT}}(q,\bar{q}) = Z_{\text{LCFT}}^0(q,\bar{q}) + \sum_{h,\bar{h}} N_{h,\bar{h}} q^h \bar{q}^{\bar{h}} \prod_{n=1}^{\infty} \frac{1}{|1-q^n|^2}, \tag{3}$$

with

$$Z_{\text{LCFT}}^0(q,\bar{q}) = Z_\Omega + Z_t = \prod_{n=2}^{\infty} \frac{1}{|1-q^n|^2} \left(1 + \frac{q^2}{|1-q|^2}\right), \tag{4}$$

where $\Omega$ is the vacuum of the holomorphic sector, and $t$ denotes the logarithmic partner of the energy momentum tensor $T$.

In [17], it was shown that the partition function of critical cosmological TMG can be expressed in terms of Bell polynomials. Writing Eq. (2) as

$$Z_{\text{cTMG}}(q,\bar{q}) = Z_{\text{gravity}}(q,\bar{q}) \cdot Z_{\text{log}}(q,\bar{q}), \tag{5}$$

where

$$Z_{\text{gravity}}(q,\bar{q}) = \prod_{n=2}^{\infty} \frac{1}{|1-q^n|^2}, \quad \text{and} \quad Z_{\text{log}}(q,\bar{q}) = \prod_{m=2}^{\infty} \prod_{\bar{m}=0}^{\infty} \frac{1}{1-q^m\bar{q}^{\bar{m}}}, \tag{6}$$

it was shown that $Z_{\text{log}}(q,\bar{q})$ is the generating function of Bell polynomials

$$Z_{\text{log}}(q,\bar{q}) = \sum_{n=0}^{\infty} \frac{Y_n}{n!} \left(q^2\right)^n, \tag{7}$$

where $Y_n$ is the (complete) Bell polynomial with variables $g_1, g_2, \ldots, g_n$ such that

$$Y_n(g_1, g_2, \ldots, g_n) = \sum_{\vec{k} \vdash n} \frac{n!}{k_1! \cdots k_n!} \prod_{j=1}^{n} \left(\frac{g_j}{j!}\right)^{k_j}, \tag{8}$$

with

$$\vec{k} \vdash n = \left\{ \vec{k} = (k_1, k_2, \ldots, k_n) \mid k_1 + 2k_2 + 3k_3 + \cdots + nk_n = n \right\}, \tag{9}$$

and

$$g_n = (n-1)! \sum_{m \geq 0, \bar{m} \geq 0} q^{nm} \bar{q}^{n\bar{m}} = (n-1)! \frac{1}{|1-q^n|^2}. \tag{10}$$

Under the rescaling of variables

$$g_n(q, \bar{q}) = (n-1)! \mathcal{G}_n(q, \bar{q}), \tag{11}$$

where

$$\mathcal{G}_n(q, \bar{q}) = \frac{1}{|1 - q^n|^2}, \tag{12}$$

the log-partition function function can also be expressed in terms of the bosonic plethystic exponential $PE^{\mathcal{B}}$ as

$$Z_{\log}(q, \bar{q}) = PE^{\mathcal{B}}[\mathcal{G}_1(q, \bar{q})] = \exp\left(\sum_{n=1}^{\infty} \frac{(q^2)^n}{n} \mathcal{G}_n(q, \bar{q})\right). \tag{13}$$

A better understanding of the partition function from a holographic perspective is desirable, in particular how to precisely match the combinatorics of multi particle logarithmic states on the gravity side to states on the (L)CFT side [18]. In order to give the aforementioned combinatorial underpinning to the non-unitary $AdS_3/LCFT_2$ holographic prescription, a modus operandi has been to study the nature of the logarithmic fields by algebro-geometric means [17,19]. This paper would like to contribute to that endeavor.

This paper is organized as follows. In section 2, we discuss integrable aspects of critical topologically massive gravity. On one hand, the partition function of the logarithmic sector of the theory is shown to be a generating function of the Burgers hierarchy in its potential form. The Burgers hierarchy is a family of nonlinear evolution equations possessing infinitely many symmetries [20], and whose simplest member is the Burgers equation, originally used to describe the mathematical modelling of turbulence [21]. Through the Hopf-Cole transformation independently introduced in [22] and [23], the Burgers equation can be converted into a linear heat equation. Conversely, starting from a hierarchy of the linear heat equation, the potential Burgers hierarchy can be defined in terms of Bell polynomials, which appear in the partition function of the logarithmic sector. We also show that the logarithmic contribution of the partition function is a $\tau$-function of an infinite (countable) family of partial differential equations known as the Kadomtsev-Petviashvili (KP) hierarchy. Historically studied as soliton solutions of KP equations, it was subsequently found that partition functions of some field theories are also solutions of the KP hierarchy. A celebrated example of the important role of $\tau$-functions of integrable hierarchies in theoretical physics is the Kontsevich model, which demonstrated in [24] that a generating function of certain intersection numbers of $\psi$-classes is a partition function of 2d topological gravity. Kontsevitch furthermore proved the conjecture proposed by Witten [25], that it coincides with a partition function of physical 2d quantum gravity, by showing that it is a $\tau$-function of the KdV hierarchy, which is a reduction of the KP hierarchy. Another example that has recently been the subject of much attention is the Jackiw-Teitelboim (JT) gravity, which describes 2d dilaton gravity and is dual to Hermitian matrix model [26]. In our work, it is shown how the 1-part Schur polynomials generated by the partition function [19] can be parametrized by certain decomposition of Grassmannian spaces in the Sato theory [27] and solve equations of the KP hierarchy. These exponential solutions are KP solitons, and display the solitonic nature the logarithmic fields. In section 3, we show that $Z_{log}(q, \bar{q})$ is also a generating function of some numerical invariants called Hurwitz numbers. Introduced at the end of the nineteenth century [28], Hurwits numbers enumerate ramified coverings of Riemann surfaces. They have appeared in various guises, from their

original treatment by means of characters of symmetric groups to applications related to the description of geometric properties in the moduli spaces of Riemann surfaces, and account for the structure of singular loci in these moduli spaces. We derive an explicit expression of the partition function in terms of Hurwitz numbers from the observation that the coefficients of the partial Bell polynomials associated to set partitions are closely related to the Hurwitz numbers associated to permutations. These results are guided by the idea of making contact with a holographic interpretation of the partition function, and motivated by an approach to analyze fields in theories on Riemann surfaces, by considering a multivalued theory on a complex plane. Such an approach consists in noticing that a Riemann surface can be represented as a branch covering of a complex plane, and visualize the Riemann surface as $n$ copies of the complex plane, "glued" at some branch points. In the considered multivalued theory, one can then construct "branch point fields" located at the projection of the branch points on the plane. Such fields first appeared in the literature in works based on the reformulation of orbifold two-dimensional conformal field theory on a Riemann surface as a theory on a branched covering of the complex plane [29–31]. In our case, the information about the geometry of the Riemann surface encoded in its monodromy properties is captured by the partition function via the Hurwitz numbers that enumerate covering spaces and leads us in section 4 to interpret the critical point $\mu l = 1$ as a branch point with the logarithmic field $\psi_{\mu\nu}^{new}$ as a branch point field associated to it. We give a summary and outlook in section 5.

## 2 Integrable hierarchies

Integrable systems have been an outstanding source of investigation in theoretical physics and mathematics, providing a wide array of applications [32,33]. They have been found essential for the description of a variety of fundamental phenomena ranging from the hydrodynamics of D=2 (space-time dimensional) non-linear soliton waves [34] and statistical mechanics [35] to string and membrane theories in high energy physics [36–38]. At a more formal level, the theory of integrable systems is a cornerstone of various basic disciplines in mathematics itself, such as algebraic geometry [39, 40], quantum groups [41, 42] or topology with link polynomials and knot theory [43].

### 2.1 Burgers hierarchy

In the last decades, several connections between combinatorics and the theory of integrable systems were discovered [44–46]. In particular, owing to recurrence relations governing the higher symmetries and conservation laws of integrable hierarchies, it is known that equations of integrable systems exhibit certain combinatorial structures. In what follows, we show that the Hilbert series generated by the logarithmic fields partition function, and studied in [17,19] are related to the Burgers and KP hierarchies.

#### 2.1.1 Potential Burgers hierarchy

The Potential Burgers hierarchy is a family of nonlinear partial differential equations (NLPDE) that features both integrable and combinatorial properties. Although the combinatorial properties of the Burgers hierarchy have been known for some time [47,48], a recent interest related to the set partition formulation of the potential form of the hierarchy has appeared in [49,50].

The connection between integrability and combinatorics can be shown from the fact that the potential Burgers hierarchy of NLPDE is logarithmically linearizable. Starting from the linear heat equation hierarchy

$$\phi_{t_n} = \phi_n, \qquad \text{with} \quad \phi_{t_n} = \frac{\partial \phi}{\partial t_n}, \quad \phi_n = \frac{\partial^n \phi}{\partial x^n}, \quad \phi \equiv \phi(t_n, x), \quad n = 0, 1, 2, \ldots, \tag{14}$$

then denoting $v_n = D^n(g)$, $D = \partial/\partial x$, and $g_{t_n} = \partial g/\partial t_n$, the change of dependent variable $\phi = e^g$ yields the potential Burgers hierarchy

$$g_{t_n} = e^{-g} D^n(e^g) = (D + g_1)^n(1) = Y_n(g_1, \ldots, g_n), \qquad n = 0, 1, 2, \ldots. \tag{15}$$

The right side of Eq. (15) displays the combinatorial aspect of the hierarchy as the multi-variable polynomials appearing are the Bell polynomials. In accordance with [17], expressing the differential operator $D$ as

$$D = \sum_{n=1}^{\infty} g_{n+1} \frac{\partial}{\partial g_n}, \tag{16}$$

it appears that $Z_{\log}(q, \bar{q})$ is a generating function of the potential Burgers hierarchy. Indeed, an infinite series of higher order symmetries -the flows- of the potential Burgers equation, can be constructed by acting the partial differential operator $D$ on the Bell polynomials, and one can see that the flows of the hierarchy are related to the partition function by this differential algebraic construction.

### 2.1.2 Burgers-Hopf hierarchy

Closely related to the Burgers hierarchy in its potential form, the Burgers-Hopf (BH) hierarchy is the system of nonlinear equations for $w = w(x, t)$ defined by

$$\partial_{t_n} w = \partial_x F_n(w), \tag{17a}$$

$$F_{n+1}(w) = (\partial_x + w)F_n(w), \qquad n = 0, 1, 2, \ldots, \qquad F_0(w) = 1, \tag{17b}$$

where $F_n(w)$ are the Faà di Bruno polynomials. The BH hierarchy, which can be obtained as a dispersionless limit of the Korteweg-de Vries (KdV) hierarchy, plays a significant role in various branches of physics and applied mathematics such as hydrodynamics or topological field theory [34, 51, 52]. The Faà di Bruno polynomials in the BH hierarchy are closely related to the Bell polynomials in the potential Burgers hierarchy [53]. The relationship can be made explicit in both forms of Burgers hierarchy by noticing that setting $w = cg_x$ and then integrating once, one introduces a dimensionless potential variable $g$ ($c$ is a dimensionless parameter) that upon appropriate choice of $c$ maps the BH to the potential Burgers hierarchy, and relates the recurrence relation in (17b) to

$$Y_{n+1} = (D + g_1)Y_n, \qquad n = 0, 1, 2, \ldots, \qquad Y_0 = 1, \tag{18}$$

that also appears in [17].

A remarkable connection between the Burgers hierarchy and the Kadomtsev-Petviashvili (KP) hierarchy was established in [54, 55], thereafter appearing in other places [56, 57]. It was shown that in the Sato theory [27], the BH is a subhierarchy of the KP hierarchy. This implies that the Burgers hierarchy solves the KP equation.

## 2.2 KP hierarchy and tau functions

In 1834, while conducting experiments to determine the most efficient design for canal boats, John Scott Russell observed a large solitary wave in a shallow water channel in Scotland [58]. The phenomenon he described as the wave of translation is now known as an example of a soliton, and is characterized by a solution of the KdV equation [59, 60], a nonlinear partial differential equation describing one-dimensional wave propagation. In 1970, Kadomtsev-Petviashvili [61] proposed a two-dimensional dispersive wave equation to study the stability of one-soliton solution of the KdV equation under the influence of weak transverse perturbations [57]. This equation is now known as the KP equation and is expressed as

$$\frac{\partial}{\partial x}\left(-4\frac{\partial f}{\partial t} + 6f\frac{\partial f}{\partial x} + \frac{\partial^3 f}{\partial x^3}\right) + 3\beta\frac{\partial^2 f}{\partial y^2} = 0, \tag{19}$$

where the function $f = f(x, y, t)$ is the wave amplitude at the point $(x, y)$ in the $xy$-plane for fixed time $t$ [33]. When $\beta = 1$, Eq. (19) is referred to as the KP II equation, and when $\beta = -1$ it is referred to as the KP I equation. With a remarkably rich structure, the KP equation is considered as the most fundamental integrable system with respect to the fact that several known integrable systems can be reformulated as special reductions of the set of the KP equation (19) together with its infinitely higher order symmetries, called the KP hierarchy.

### 2.2.1 The KP hierarchy

The KP hierarchy is a completely integrable system of quadratic partial differential equations for a function $F(\mathcal{G}_1, \mathcal{G}_2, \ldots)$ depending on infinitely many variables. Each equation is indexed by partitions of integers $n \geq 4$, and can be organized in such a way that its left hand side corresponds to partitions into two parts none of which is 1, while its right hand side corresponds to partitions of the same number $n$ involving 1 [62]. The lowest KP hierarchy equation then corresponds to the partition of 4 expressed as

$$\frac{\partial^2 F}{\partial \mathcal{G}_2^2} = \frac{\partial^2 F}{\partial \mathcal{G}_1 \partial \mathcal{G}_3} - \frac{1}{2}\left(\frac{\partial^2 F}{\partial \mathcal{G}_1^2}\right)^2 - \frac{1}{12}\frac{\partial^4 F}{\partial \mathcal{G}_1^4}, \tag{20}$$

or in a more compact form as

$$F_{2^1 2^1} = F_{1^1 3^1} - \frac{1}{2}\left(F_{1^1 1^1}\right)^2 - \frac{1}{12}F_{1^4}, \tag{21}$$

where [63]

$$F_{1^{a_1}\cdots r^{a_r}} = \frac{\partial^{a_1 + \cdots + a_r}}{\partial \mathcal{G}_1^{a_1}\cdots \partial \mathcal{G}_r^{a_r}}F. \tag{22}$$

While the second equation of the KP hierarchy, corresponding to the partition of 5 can be expressed as

$$F_{2^1 3^1} = F_{1^1 4^1} - \frac{1}{6}F_{1^3 2^1} - F_{1^2}F_{1^1 2^1}, \tag{23}$$

there are two equations at the third level, i.e for $n = 6$, corresponding to the partitions $2 + 4 = 6$ and $3 + 3 = 6$. At each level $n$, the number of equations increases according to the appropriate partitions of $n$ as outlined above.

A striking development in the KP theory occurred in 1981, when by means of algebraic analysis methods, Sato [27] realized that the solutions of the KP hierarchy could be written in terms of points representing a $GL_\infty$-orbit on an infinite dimensional Grassmann manifold, which he called the Universal Grassmann Manifold (UGM). This remarkable link between the theory of solitons and the group $GL_\infty$ constitutes a unified approach to integrability commonly known as the Sato theory, and provides a deep insight into algebraic and geometric properties of integrable systems with infinitely many degrees of freedom.

### 2.2.2 Sato theory

In Sato theory, a system of soliton equations is defined as a dynamical system on an infinite dimensional Grassmannian by interpreting the space of solutions of the KP hierarchy as the Grassmannian of semi-infinite planes in an infinite dimensional vector space. At the heart of this geometric construction lies the idea that under a projective embedding, the KP hierarchy is expressed as algebraic identities of the Grassmannian called Plücker relations, in terms of a coordinate system called the Plücker coordinates. Each solution of the KP hierarchy associated to a point of the Grassmaniann is given by a function of infinitely many variables called the $\tau$-function, which is related to the Plücker embedding. A brief exposé of this correspondence is given below.

**Projective embedding**
Consider the $n$-dimensional complex vector space $V \simeq \mathbb{C}^n$. For $0 < k < n$, the $k^{\text{th}}$ wedge product space

$$\bigwedge^k V = \{v_{\mu_1} \wedge v_{\mu_2} \wedge \cdots \wedge v_{\mu_k}, \quad v_{\mu_j} \in V\} \tag{24}$$

is a vector space of dimension $\binom{n}{k}$. Let $W \subset V$ be a $k$-dimensional vector subspace of $V$ with basis $\{w_1, w_2, \ldots, w_k\}$, such that an element

$$w_1 \wedge w_2 \wedge \cdots \wedge w_k \in \bigwedge^k V \tag{25}$$

can be assigned to each basis of $W$. The Grassmannian $Gr(k, V)$ is the set of all $k$-dimensional vector subspaces of $V$, which has the structure of a smooth compact algebraic variety of dimension $k(n-k)$. The assignment (25) defines the Plücker embedding

$$Gr(k, V) \longrightarrow \mathbb{P}\left(\bigwedge^k V\right) \tag{26}$$

of the Grassmannian into the projective space $\bigwedge^k V$. To understand this, we can consider the example of the finite dimensional Grassmannian $Gr(2, 4)$, which is the set of all 2-dimensional planes passing through the origin of a 4-dimensional vector space $V \equiv \mathbb{C}^4$ [64]. To introduce a coordinate system on this Grassmannian, one can take $v_i$ ($i = 1, \ldots, 4$) as a vector basis for the 4-dimensional vector space $V \equiv \mathbb{C}^4$, and express a basis $\{w_1, w_2\}$ for any 2-dimensional plane in $V$ by means of the coordinates $a_{ij}$ of the $w_j$ in the $\{v_i\}$ basis

$$w_i = \sum_{j=1}^4 a_{ji} v_j. \tag{27}$$

Denoting the manifold of all $4 \times 2$ matrices of rank 2 by $M(4,2)$, the matrix $\left(a_{ij}\right)_{4\times 2} \in M(4,2)$ may be defined as the frame of $W$ in $V$, and one can think of the Grasmannian $Gr(2,4)$ as the quotient space obtained by action of $GL(2,\mathbb{C})$ on $M(4,2)$. Then, using the minor determinant of the frame $\left(a_{ij}\right)_{4\times 2}$, under the embedding

$$Gr(2,V) \longrightarrow \mathbb{P}\left(\bigwedge^2 V\right), \qquad V \equiv \mathbb{C}^4, \tag{28a}$$

$$(w_1, w_2) \longrightarrow w_1 \wedge w_2, \tag{28b}$$

one can define the following homogeneous coordinates in 5-dimensional projective space $\mathbb{P}^5$

$$\mathcal{A} = \left(A_{12}, A_{13}, A_{14}, A_{23}, A_{24}, A_{34}\right) \tag{29a}$$

$$= \left(\begin{vmatrix} a_{11} & a_{12} \\ a_{21} & a_{22} \end{vmatrix}, \begin{vmatrix} a_{11} & a_{12} \\ a_{31} & a_{32} \end{vmatrix}, \begin{vmatrix} a_{11} & a_{12} \\ a_{41} & a_{42} \end{vmatrix}, \begin{vmatrix} a_{21} & a_{22} \\ a_{31} & a_{32} \end{vmatrix}, \begin{vmatrix} a_{21} & a_{22} \\ a_{41} & a_{42} \end{vmatrix}, \begin{vmatrix} a_{31} & a_{32} \\ a_{41} & a_{42} \end{vmatrix}\right). \tag{29b}$$

The action of $GL(2,\mathbb{C})$ on $w_1, w_2$ only results in the multiplication of each minor by the determinant of the transformation matrix, and because $\mathcal{A}$ is invariant under such transformations, $Gr(2,4)$ can therefore be regarded as a subvariety of $\mathbb{P}^5$. In other words, choosing another basis $\{u_1, u_2\}$ of $W$, there is a $2 \times 2$ matrix $M \in GL(2,\mathbb{C})$ such that

$$(u_1, u_2) = M \cdot (w_1, w_2). \tag{30}$$

Then

$$u_1 \wedge u_2 = \det(M) \cdot w_1 \wedge w_2, \tag{31}$$

under the Plücker embedding. Thus $u_1 \wedge u_2$ and $w_1 \wedge w_2$ belong to the same line in the vector space $\left(\bigwedge^2 V\right)$, which therefore defines the same element of the projective space $\mathbb{P}\left(\bigwedge^2 V\right)$. Taking $Gr(k,V)$ as the quotient space obtained by the right-action of the Lie group $GL(k,\mathbb{C})$ on the manifold $M(m,n)$ of all $n \times m$ matrices of rank $m$, the above procedure can be generalized to the embedding of any $k(n-k)$-dimensional Grassmannian of $k$-dimensional planes in $n$-dimensional vector spaces $V$ in the projectivization of $\mathbb{P}\left(\bigwedge^k V\right)$ using the Plücker coordinates. These coordinates are not independent, but are rather fully characterized by a set of nonlinear algebraic relations called Plücker relations. In the case of $Gr(2,4)$ considered above, $\mathcal{A}$ is the Plücker coordinate, and it satisfies only one relation, which can be obtained by expansion of the zero determinant

$$\begin{vmatrix} a_{11} & a_{12} & 0 & 0 \\ a_{21} & a_{22} & a_{21} & a_{22} \\ a_{31} & a_{32} & a_{31} & a_{32} \\ a_{41} & a_{42} & a_{41} & a_{42} \end{vmatrix} = a_{12}a_{34} - a_{13}a_{24} + a_{14}a_{23} = 0. \tag{32}$$

For arbitrary choices of $n$ and $k$, the Plücker equations remain quadratic.

**Infinite dimensional vector space of Laurent polynomials and its semi-infinite wedge product space**

In order to have an infinite-dimensional generalization, consider the ring of Laurent polynomials in one variable $V = \mathbb{C}[z, z^{-1}]$, as the vector space with basis $z^j, j \in \mathbb{Z}$ and elements of the form

$$t_{-k} z^{-k} + t_{-k+1} z^{-k+1} + \cdots . \tag{33}$$

By definition, the semi-infinite wedge product space

$$\bigwedge^{\infty/2} V = \{v_\lambda : v_\lambda = z^{l_1} \wedge z^{l_2} \wedge v_\lambda = z^{l_3} \wedge \cdots , \quad l_j = \lambda_j - j\} \tag{34}$$

is a vector space with basis vectors $v_\lambda$ freely spanned by the infinite wedge products where $\lambda$ is a partition, $\lambda = (\lambda_1, \lambda_2, \lambda_3, \ldots)$, $\lambda_1 \geq \lambda_2 \geq \lambda_3 \geq \ldots$, having all but finitely many parts equal to 0. In particular $l_j = -j$ for $j$ large enough. In such a vector space, while for instance the lowest vector, corresponding to the empty partition reads

$$v_\emptyset = z^{-1} \wedge z^{-2} \wedge z^{-3} \wedge \cdots , \tag{35}$$

the next three vectors are expressed as

$$
\begin{aligned}
v_{1^1} &= z^0 \wedge z^{-2} \wedge z^{-3} \wedge \cdots , \\
v_{2^1} &= z^1 \wedge z^{-2} \wedge z^{-3} \wedge \cdots , \\
v_{1^2} &= z^0 \wedge z^{-1} \wedge z^{-3} \wedge \cdots .
\end{aligned}
\tag{36}
$$

**Boson-fermion correspondence**

The semi-infinite wedge power $\bigwedge^{\infty/2} V$, also called Fermionic Fock space $\mathcal{F}$ is an irreducible representation of the Clifford algebra that contains the Lie algebras $gl\left(\bigwedge^k V\right)$, for all $k \geq 0$. In particular, the Fock space $\mathcal{F}$ is a $gl\left(\bigwedge^k V\right)$-module, for all $k \geq 0$.

As a vector space, the bosonic Fock space $\mathcal{B}$ is a polynomial ring in countably many variables. Via a natural module isomorphism over the infinite dimensional Lie Heisenberg algebra called the boson-fermion correspondence, the bosonic Fock space $\mathcal{B} = \mathbb{C}[\mathcal{G}_1, \mathcal{G}_2, \ldots]$ can also be endowed a $gl\left(\bigwedge^k V\right)$-module structure. This isomorphism $\mathcal{B} \longleftrightarrow \mathcal{F}$ takes the basis vector $v_\lambda$ to the Schur polynomials $\chi_R(\mathcal{G}_1, \mathcal{G}_2, \ldots)$, that is the character-polynomials associated to the irreducible tensor representation of the GL group classified in terms of Young diagrams $R$ [19].

The Schur polynomials $\chi_R(\mathcal{G}_1, \mathcal{G}_2, \ldots)$ parametrized by Young diagrams $R = \overbrace{\square \cdots \square}^{k}$ corresponding to the one-part partition $k^1 \vdash k$ can be defined in terms of the power series expansion

$$\chi_\emptyset + \chi_\square z + \chi_{\square\square} z^2 + \chi_{\square\square\square} z^3 + \cdots = \exp\left\{\mathcal{G}_1 z + \mathcal{G}_2 \frac{z^2}{2} + \mathcal{G}_3 \frac{z^3}{3} + \cdots\right\} \tag{37a}$$

$$= 1 + \mathcal{G}_1 z + \frac{1}{2}\left(\mathcal{G}_1^2 + \mathcal{G}_2\right) z^2 + \frac{1}{6}\left(\mathcal{G}_1^3 + 3\mathcal{G}_1 \mathcal{G}_2 + 2\mathcal{G}_3\right) z^3 + \cdots . \tag{37b}$$

Young diagrams are graphical representations of integer partitions, hence for an arbitrary $\lambda = (\lambda_1, \lambda_2, \ldots, \lambda_k)$, $\lambda_1 \geq \lambda_2 \geq \cdots \geq \lambda_k$, the Schur polynomial $\chi_R \equiv \chi_\lambda$ can be obtained through the Jacobi-Trudi formula

$$\chi_\lambda = \det \begin{pmatrix} \chi_{\lambda_1} & \chi_{\lambda_1+1} & \chi_{\lambda_1+2} & \cdots & \chi_{\lambda_1+k-1} \\ \chi_{\lambda_2-1} & \chi_{\lambda_2} & \chi_{\lambda_2+1} & \cdots & \chi_{\lambda_2+k-2} \\ \cdots & \cdots & \cdots & \cdots & \cdots \\ \chi_{\lambda_k-k+1} & \chi_{\lambda_k-k+2} & \chi_{\lambda_k-k+3} & \cdots & \chi_{\lambda_k} \end{pmatrix}. \tag{38}$$

Some of the first few Schur polynomials are expressed as follows

$$\chi_\emptyset = 1, \quad \chi_\square = \mathcal{G}_1, \quad \chi_{\square\square} = \tfrac{1}{2}\left(\mathcal{G}_1^2 + \mathcal{G}_2\right), \quad \chi_{\square\square\square} = \tfrac{1}{6}\left(\mathcal{G}_1^3 + 3\mathcal{G}_1\mathcal{G}_2 + 2\mathcal{G}_3\right),$$

$$\chi_{\boxminus} = \tfrac{1}{2}\left(\mathcal{G}_1^2 - \mathcal{G}_2\right), \quad \chi_{\boxminus} = \tfrac{1}{3}\left(\mathcal{G}_1^3 - \mathcal{G}_3\right), \quad \chi_{\boxminus} = \tfrac{1}{6}\left(\mathcal{G}_1^3 - 3\mathcal{G}_1\mathcal{G}_2 + 2\mathcal{G}_3\right).$$

**Geometric representation of the KP hierarchy**

Let $\Gamma \subset V$ be a "half-infinite dimensional vector subspace" of $V$ with basis $\{\gamma_1, \gamma_2, \gamma_3, \ldots\}$, such that an element

$$\gamma_1 \wedge \gamma_2 \wedge \gamma_3 \wedge \cdots \in \bigwedge^{\infty/2} V \tag{39}$$

can be assigned to each basis of $\Gamma$. Then, the Grassmannian $Gr(\infty/2, V)$ is the set of all half-infinite dimensional vector subspaces of $V$, and the assignment (39) defines the Plücker embedding

$$Gr(\infty/2, V) \longrightarrow \mathbb{P}\left(\bigwedge^{\infty/2} V\right) \tag{40}$$

of the *Sato Grassmannian* into the projective space $\bigwedge^{\infty/2} V$. The vector space $Gr(\infty/2, V)$ can be interpreted as the wedge product

$$\gamma_1(z) \wedge \gamma_2(z) \wedge \gamma_3(z) \wedge \cdots, \tag{41}$$

where $\gamma_i$ is a Laurent power series in $z$ such that for sufficiently large $i$, $\gamma_i = z^{-i} +$ terms of higher order in $z$. Then through the boson-fermion coordinate isomorphism of vector spaces

$$\mathbb{C}[\mathcal{G}_1, \mathcal{G}_2, \ldots] \simeq \bigwedge^{\infty/2} V, \tag{42}$$

the function $\tau \in \mathbb{C}[\mathcal{G}_1, \mathcal{G}_2, \ldots]$ represented by the wedge product $\tau \longleftrightarrow \gamma_1 \wedge \gamma_2 \wedge \gamma_3 \wedge \cdots$ is the exponent of a solution of the KP hierarchy.

By definition, the Plücker embedding of the Grassmannian is given by quadratic equations called *Hirota bilinear equations*. These algebraic equations can be expressed in terms of the logarithm $F = \log \tau$ as partial differential equations that are exactly the equations of the KP hierarchy. Hence, the $\tau$ function of the KP hierarchy is a solution of the Hirota equations.

### 2.2.3 Logarithmic partition function as a KP $\tau$-function

The Schur polynomials give rational solutions of the KP hierarchy, and form a convenient language to describe $\tau$-functions from the fact that [62], any linear combination

$$1 + k_1 \chi_\square + k_2 \chi_{\square\square} + k_3 \chi_{\square\square\square} + \cdots \tag{43}$$

of one-part Schur polynomials with constant coefficients $k_n$ is a $\tau$- function of the KP hierarchy. Hence

$$Z_{log}(q, \bar{q}) = \sum_{n=0}^{\infty} \chi_{\underbrace{\square \ldots \square}_{n}} \cdot (q^2)^n \tag{44}$$

is a KP $\tau$-function.

## 3 Critical TMG Hurwitz partition function

A Hurwitz number counts the number of non-equivalent branched coverings of a surface with a prescribed set of branch points and branched profile. Although branched coverings first appeared in [65], their enumeration was studied in a systematic way by Hurwitz who observed that the counting of branched coverings could be interpreted in terms of permutation factorizations [28,66]. Ever since, Hurwitz numbers have been an important subject in mathematics and physics, with a copious amount of literature dedicated to them [67–72, 72–74]. They have been found notably in the context of string theory after a crucial observation made in [75,76] from which many works followed, in integrable systems with early works [77,78], or in matrix models [79–83].

It was observed that the generating function of a particular type of Hurwitz numbers called double Hurwitz numbers is a $\tau$-function of the 2D Toda hierarchy [77]. This generating function can be specialized to a generating function of single Hurwitz numbers, which becomes a $\tau$-function of the KP hierarchy [63, 84, 85]. In this section, we will show that $Z_{log}(q, \bar{q})$ which was shown to be a KP $\tau$-function in the previous section is also a generating function of (single) Hurwitz numbers.

### 3.1 Preliminary definitions

We start with brief preliminary definitions of ramifications on branched covering maps and Hurwitz numbers. Our definitions and notation follow [86].

**Ramification** Let $f : X \mapsto Y$ be a holomorphic map of Riemann surfaces $X, Y$ of degree $n$, $y \in Y$, and let $f^{-1}(y) = \{x_1, \ldots, x_d\}$. From complex analysis, holomorphic maps can be given a local expression of the form $z \mapsto z^k$, with $k \in \mathbb{Z}^+$. When the local expression for any $x_i \in f^{-1}(y)$ is $w \mapsto w^{k_{x_i}}$, the positive integer $k_{x_i}$ is called the ramification index (or order) of $f$ at $x_i$, and the set $k_{x_1}, \ldots, k_{x_d}$ is the ramification profile of $f$ at $y$. Note that the ramification profile of $f$ at $y$ is a partition of $n$.

**Covering maps** A surjective continuous map $f : X \mapsto Y$ is called a covering map if for every $y \in Y$ and each $x_i$ in a discrete set $f^{-1}(y)$, there exists a neighborhood $U \subset Y$ such that the preimage $f^{-1}(U) \subset X$ consists of disjoints neighborhoods $V_{x_i}$ and each restriction of $f$ is to $V_{x_i}$ is a homeomorphism $f : V_{x_i} \mapsto U$. Intuitively, such a map is a trivial cover in the sense that $X$

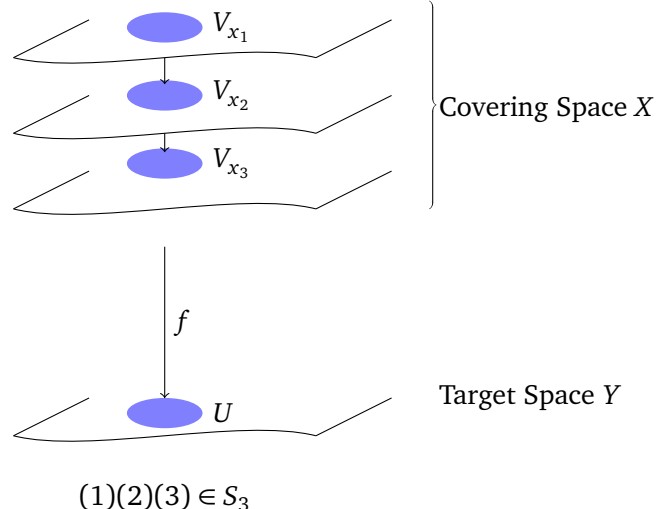

Figure 1: A 3-fold covering space $X$ over a target space $Y$.

simply consists of a number of disjoint copies of $Y$, and the map restricted to each copy is just the identity function. As we will see, what is meant by function here is in fact permutation. This definition is illustrated in Fig. 1. The copies of the target space $Y$ are the sheets of the covering space $X$

**Branched coverings**   A surjective continuous map $f : X \mapsto Y$ is called a branched covering map if there is a finite set of points $B \subset Y$ such that $f^{-1}(B) \subset X$ is finite and $f : X \setminus f^{-1}(B) \mapsto Y \setminus B$ is a covering. The set $B$ is called branch locus of $f$, and the points $y_j$ in $B$ are called branch points of $f$ [87].

A branched covering is a generalization of a covering space in which the sheets are allowed to "collide" over a locus in the target space. This is illustrated in Fig. 2 in which we think of a branched covering as a covering space where some of the sheets are identified over the branch locus.

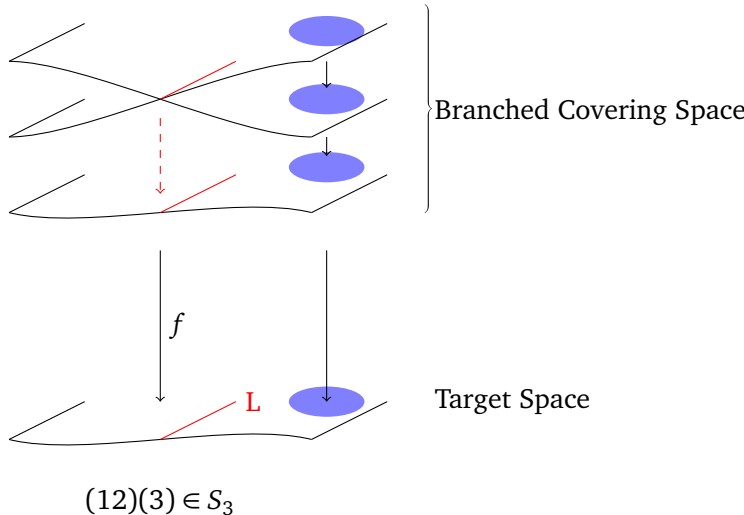

Figure 2: A 3-fold branched covering $X$, where the branch locus is the line $L$ in $Y$.

**Hurwitz numbers** Let $Y$ be a connected Riemann surface of genus $g$. Define the set $B = \{y_1,\ldots,y_d\} \in Y$, and let $\lambda_1,\ldots,\lambda_d$ be partitions of the positive integer $n$. Then the Hurwitz number can be defined as the sum

$$H_{X \xrightarrow{n} Y}(\lambda_1,\ldots,\lambda_d) = \sum_{|f|} \frac{1}{|\mathrm{Aut}(f)|} \tag{45}$$

that runs over each isomorphism class of $f : X \mapsto Y$ where

1. $f$ is a holomorphic map of Riemann surfaces;

2. $X$ is connected and has genus $h$;

3. the branch locus of $f$ is $B = \{y_1,\ldots,y_d\}$;

4. the ramification profile of $f$ at $y_i$ is $\lambda_i$.

Hurwitz numbers come in two different types, depending on whether the covering space $X$ of $Y$ is connected or not. Before giving the definition of disconnected Hurwitz numbers, we would like to mention a few facts relevant to the study of $Z_{log}(q,\bar{q})$ as a generating function of Hurwitz numbers. Firstly, although we will need to start with the above definition of the connected Hurwitz number, our focus will be on the disconnected theory. Besides, we will restrict our attention to the target space with genus $g = 0$. In that case, the problem at hand can be further treated by using the representation theory of the symmetric group.

Let $\lambda_1,\ldots,\lambda_d$ be partitions of the positive integer $n$. Recall from the representation theory of the symmetric group that $\mathfrak{Z}(\mathbb{C}[S_n])$ is a vector space with dimension equal to the number of partitions of $n$ and basis indexed by conjugacy classes of permutations. Denoting the basis element associated to the corresponding conjugacy class by $C_{\lambda_i}$ for every $i \in [1;d]$, the genus zero disconnected Hurwitz number takes the form

$$H^{\bullet}_{X \xrightarrow{n} 0}(\lambda_1,\ldots,\lambda_d) = \frac{1}{n!}[C_e] C_{\lambda_d} \cdots C_{\lambda_2} C_{\lambda_1}, \tag{46}$$

where $[C_e] C_{\lambda_d} \cdots C_{\lambda_2} C_{\lambda_1}$ is the coefficient of $C_e = \{e\}$ after writing the product as a linear combination of the basis element $C_{\lambda_i}$.

## 3.2 Hurwitz partition function

At present, we are ready to show that the partition function of the logarithmic sector is a Hurwitz partition function. We start by a further restriction on the genus of the Riemann surface $X$ of degree $n$, by imposing that $h = 0$. In that case, letting $\lambda_1 = \lambda_2 = (n)$, the expression of connected Hurwitz numbers becomes

$$H_{0 \xrightarrow{n} 0}((n),(n)) = \frac{1}{n}. \tag{47}$$

According to the plethystic program [88], given the moduli space $\mathbb{C}^2$ of the theory, the Hilbert series

$$\mathcal{G}(q,\bar{q}) = \frac{1}{(1-q)(1-\bar{q})} \tag{48}$$

is the generating function of basic single-trace invariants of the theory. To find the multi-trace invariants, i.e the unordered products of the single-trace invariant objects, we take the plethystic exponential

$$Z(q,\bar{q}) = PE\left[\mathcal{G}(q,\bar{q})\right] = \exp\left\{\sum_{k=1}^{\infty} \frac{\mathcal{G}_k(q,\bar{q}) - \mathcal{G}(0,0)}{k}\right\} = \prod_{n,m}^{\infty} \frac{1}{1 - q^n \bar{q}^m}\,, \tag{49}$$

$$\mathcal{G}_k(q,\bar{q}) = \frac{1}{(1-q^k)(1-\bar{q}^k)}\,. \tag{50}$$

$\mathcal{G}(q,\bar{q})$ is therefore the inverse function of a plethystic exponential also called plethystic logarithm, and using Eq. (47), we can rewrite it as

$$\mathcal{G}(q,\bar{q}) = \sum_{n=1}^{\infty} \frac{\mu(n)}{n} \log\left[Z(q^n,\bar{q}^n)\right] = \sum_{n=1}^{\infty} \left[H_{0 \xrightarrow{n} 0}((n),(n))\right]\mu(n)\log\left[Z(q^n,\bar{q}^n)\right], \tag{51}$$

where $\mu(n)$ is the Möbius function

$$\mu(n) = \begin{cases} 0\,, & n \text{ has repeated prime factors,} \\ 1\,, & n = 1\,, \\ (-1)^i\,, & n \text{ is a product of } i \text{ distinct primes.} \end{cases} \tag{52}$$

The plethystic logarithm $\mathcal{G}(q,\bar{q})$ is thus a generating function of connected Hurwitz numbers. When promoting the plethystic exponential $PE\left[\mathcal{G}(q,\bar{q})\right]$ to a $(q^2)$ parameter-inserted version, we obtain the partition function

$$Z_{log}(q,\bar{q}) = \sum_{n=0}^{\infty} Z_n(q,\bar{q})\left(q^2\right)^n = \exp\left\{\sum_{k=1}^{\infty} \left[H_{0 \xrightarrow{k} 0}((k),(k))\right]\mathcal{G}_k(q,\bar{q})\left(q^2\right)^k\right\}, \tag{53}$$

where the Hilbert series of the $n$-th symmetric product is given by

$$Z_n\left(q,\bar{q};\mathbb{C}^2\right) = Z_1\left(q,\bar{q};\text{sym}^n\left(\mathbb{C}^2\right)\right) = \mathcal{G}\left(q,\bar{q};\text{sym}^n\left(\mathbb{C}^2\right)\right), \qquad \text{sym}^n\left(\mathbb{C}^2\right) = \left(\mathbb{C}^2\right)^n/S_n. \tag{54}$$

Connected and disconnected Hurwitz generating functions are related by exponentiation, hence $Z_{log}(q,\bar{q})$ is in fact a generating function of disconnected Hurwitz numbers. We now show how they appear in $Z_n(q,\bar{q})$.
Let $n = 1,2,\ldots$ and $k = 1,\ldots,n$. Moreover, let $p(n,k)$ denote the tuple of nonnegative integer solutions $j := j_1,\ldots,j_n$ of the system

$$\begin{cases} j_1 + j_2 + \cdots + j_n = k\,, \\ j_1 + 2j_2 + \cdots + nj_n = n\,. \end{cases} \tag{55}$$

The generalized binomial coefficient defined as

$$\binom{n}{j_1,\ldots,j_k} = \frac{n!}{j_1!j_2!\cdots j_n!\cdot(1!)^{j_1}(2!)^{j_2}\cdots(n!)^{j_n}}\,, \tag{56}$$

can be interpreted in terms of partitions by considering partitions of the set $\{1, 2, \ldots, n\}$ into $k$ blocks of $j_i$ $i$ elements subsets such that the system (55) holds. Then the number of all partitions of this type is equal to the binomial coefficient. The polynomial

$$B_{n,k}(g_1, g_2, \ldots) := \sum_{p(n,k)} \binom{n}{j_1, \ldots, j_k} g_1^{j_1} g_2^{j_2} \cdots g_n^{j_n} \tag{57}$$

is called the Bell polynomial of type $n, k$, or partial Bell polynomial, and the partition function of the logarithmic fields can be written as

$$Z_{log}(q, \bar{q}) = 1 + \sum_{n=1}^{\infty} \frac{1}{n!} \left( \sum_{k=1}^{n} B_{n,k}(g_1, g_2, \ldots) \right) (q^2)^n. \tag{58}$$

From the coordinate substitution $g_n = (n-1)! \mathcal{G}_n$, the partition function becomes

$$Z_{log}(q, \bar{q}) = 1 + \sum_{n=1}^{\infty} \frac{1}{n!} \left( \sum_{k=1}^{n} \frac{n!}{j_1! j_2! \cdots j_n! \cdot (1)^{j_1} (2)^{j_2} \cdots (n)^{j_n}} \mathcal{G}_1^{j_1} \mathcal{G}_2^{j_2} \cdots \mathcal{G}_n^{j_n} \right) (q^2)^n \tag{59a}$$

$$= 1 + \sum_{n=1}^{\infty} \left( \sum_{k=1}^{n} \frac{1}{j_1! j_2! \cdots j_n! \cdot (1)^{j_1} (2)^{j_2} \cdots (n)^{j_n}} \mathcal{G}_1^{j_1} \mathcal{G}_2^{j_2} \cdots \mathcal{G}_n^{j_n} \right) (q^2)^n \tag{59b}$$

$$= 1 + \sum_{n=1}^{\infty} \left( \sum_{k=1}^{n} \left[ H^{\bullet}_{0 \xrightarrow{n} 0} \left( ([1]^{j_1}, [2]^{j_2}, \ldots), ([1]^{j_1}, [2]^{j_2}, \ldots) \right) \right] \mathcal{G}_1^{j_1} \mathcal{G}_2^{j_2} \cdots \mathcal{G}_n^{j_n} \right) (q^2)^n. \tag{59c}$$

The disconnected Hurwitz number expression takes the form

$$H^{\bullet}_{0 \xrightarrow{n} 0} \left( ([1]^{j_1}, [2]^{j_2}, \ldots), ([1]^{j_1}, [2]^{j_2}, \ldots) \right) = \frac{1}{j_1! j_2! \cdots j_n! \cdot (1)^{j_1} (2)^{j_2} \cdots (n)^{j_n}}, \tag{60}$$

where the $\left([1]^{j_1}, [2]^{j_2}, \ldots\right)$ associated to the monomials $\mathcal{G}_1^{j_1} \mathcal{G}_2^{j_2} \cdots$ are such that $[i]^{j_i} = \overbrace{i, \ldots, i}^{j_i \text{ times}}$.

As an example, the set $\{1, 2, 3\}$ has five partitions. Three of these have two blocks, namely $\{1, 2\}\{3\}$, $\{1, 3\}\{2\}$ and $\{2, 3\}\{1\}$. We associate the monomial $\mathcal{G}_1 \mathcal{G}_2$ to each of them, with data $k = 2, j_1 = j_2 = 1$. Then the Hurwitz number can be computed as

$$H^{\bullet}_{0 \xrightarrow{n} 0} ((1, 2), (1, 2)) = \frac{1}{1! 1! \cdot (1)^1 (2)^1} = \frac{1}{2}. \tag{61}$$

This result can be obtained using Eq. (46) known in the literature, by considering the basis element $C_{(1,2)} = (12) + (13) + (23)$ of the class algebra $\mathfrak{Z}(\mathbb{C}[S_3])$. Then

$$[(12) + (13) + (23)][(12) + (13) + (23)] = 3e + 3(123) + 3(132) = 3C_e + 3C_{(3)}, \tag{62}$$

and

$$H^{\bullet}_{0 \xrightarrow{n} 0} ((1, 2), (1, 2)) = \frac{1}{3!} \cdot 3 = \frac{1}{2}. \tag{63}$$

Note that when $k = n$, i.e for one block partitions

$$H^{\bullet}_{0 \xrightarrow{n} 0} ((n), (n)) = H_{0 \xrightarrow{n} 0} ((n), (n)). \tag{64}$$

# 4 Branched coverings and logarithmic branch point field

It was found in [19] that the moduli space of the logarihmic fields is the Calabi-Yau orbit space $\left(\mathbb{C}^2\right)^n/S_n$. Two natural questions to ask are whether this quotient fits in with covering spaces, and if so, what more can be inferred about the theory. Branched coverings are particularly interesting when working with generic manifolds because one can construct a manifold from multiple copies of a simpler manifold sewn together in a specific way. From there, one can just calculate physical quantities on the covering spaces. As can be seen from the above work, the logarithmic partition function is an example of such a computation. Hurwitz theory studies maps between Riemann surfaces, and can be used to learn more about the geometry of the moduli spaces of the Riemann surfaces. The Hurwitz numbers in the partition function are counts of the maps between Riemann surfaces. Thus, the partition function can be regarded as the generating function of $n$-coverings of $\mathbb{C}^2$ with the symmetric group acting on it, and the covering of $\left(\mathbb{C}^2\right)^n/S_n$ is the map

$$f : \left(\mathbb{C}^2\right)^n \mapsto \left(\mathbb{C}^2\right)^n/S_n, \tag{65}$$

which allows sheets of $\left(\mathbb{C}^2\right)^n$ to collide over a locus in $\left(\mathbb{C}^2\right)^n/S_n$ (see for instance Fig. 2 for $n = 3$), associated to non trivial permutations of the sheets of the coverings. In this way, the first question has been addressed, and we now want to answer the second question.

Two-dimensional conformal field theory on a Riemann surface can be formulated as a theory on a branched covering of the complex plane, where a branch point corresponds to the insertion of a conformal primary field known as branch point twist field. The idea of quantum fields associated to branched points of Riemann surfaces first emerged in the context of orbifold conformal field theory [29–31]. Incidentally, the work of [30] also marks the first instance of a logarithmic conformal field theory in the literature.

The basic idea of the work in [30] is that a compact Riemann surface can always be represented as a $n$-fold ramified covering of the complex plane or, more precisely, its compactified version, the Riemann sphere. Then, the computation of correlation functions on the non-trivial Riemann surface was replaced by computations on the sphere through the insertions of suitable operators which simulate the branch points of a ramified covering of the sphere. As it appeared in [10], the critical point $\mu l = 1$ is a branch point. We argue that the critical point therefore corresponds to the insertion of a field, $\psi_{\mu\nu}^{new}$, and we discuss the meaning of the new field as a branch point field.

The interpretation of $\psi_{\mu\nu}^{new}$ as a logarithmic branch point field is due to the presence of the complex logarithmic function $y(\tau, \rho) = -\ln \cosh \rho - i\tau$ in the wave function defined in [10] as

$$\psi_{\mu\nu}^{new} := \lim_{\mu l \to 1} \frac{\psi_{\mu l}^M(\mu l) - \psi_{\mu l}^L}{\mu l - 1} = y(\tau, \rho)\psi_{\mu l}^L. \tag{66}$$

The complex logarithm $y(\tau, \rho) = -\ln \cosh \rho - i\tau$ can be defined as the inverse of the exponential function satisfying $e^{-\ln \cosh \rho - i\tau} = z$, such that $y(\tau, \rho) = \log z$. Since $e^{-\ln \cosh \rho - i\tau + 2\pi i} = e^{-\ln \cosh \rho - i\tau}$, $z$ is periodic and there are infinitely many ways to define $y(\tau, \rho) = -\ln \cosh \rho - i\tau$, which differ by multiples of $2\pi i$. $y$ can therefore be written as

$$y(\tau, \rho) = -\ln \cosh \rho - i\tau + i(2\pi n), \qquad n \in \mathbb{Z}. \tag{67}$$

Every time $z$ moves in a closed curve around the branch point $\mu l = 1$, $\tau$ increases by $2\pi$ and we go from one branch of the complex logarithm to another. Any branch of the logarithm is

discontinuous along its branch cut. The complex logarithm $y(\tau, \rho)$ is therefore an infinitely multivalued function, continuous only on branches of the logarithm separated from each other by multiples of $2\pi i$, precisely by branch cuts ending at the branch points $\mu l = 1$ and infinity. Hence, $\psi_{\mu\nu}^{new}$ is singlevalued on infinitely many copies of $\mathbb{C}^2$ cut along a locus, stacked directly upon each other, and connected along the branch cuts.

Putting our results together, i.e

1. $Z_{log}(q, \bar{q})$ is a generating function of the potential Burger's hierarchy (Eq. (15)),

2. $Z_{log}(q, \bar{q})$ is a $\tau$-function of the KP hierarchy,

3. $Z_{log}(q, \bar{q})$ is a Hurwitz partition function (Eq. (59c)),

4. $\psi_{\mu\nu}^{new}$ is a branch point field associated to the branch point $\mu l = 1$,

quite remarkably, the integrability properties of the partition function of the logarithmic sector of TMG at the critical point bring new perspectives to the theory. The connection between integrability and Hurwitz numbers reveals the branched covering picture of a Riemann surface in the geometry of the theory, which in turn allows an interpretation of the appearance of the logarithmic partner of $\psi_{\mu\nu}^{L}$ in [10] as the insertion of a branch point field $\psi_{\mu\nu}^{new}$ at the branch point $\mu l = 1$ of the Riemann surface. In addition to the known fact that the complex logarithmic function $y(\tau, \rho) = -\ln\cosh\rho - i\tau$ is responsible for the feature of a Jordan cell involving $\psi_{\mu\nu}^{new}$ and $\psi_{\mu\nu}^{L}$ [10], a novel feature appears in the fact that because of its inbuilt function $y(\tau, \rho)$, the branch point field $\psi_{\mu\nu}^{new}$ located at the projection of the branch point $\mu l = 1$ simulates the effects of the geometry of the non-trivial Riemann surface. These results constitute an interesting point d' appui for further investigations.

## 5 Summary and outlook

In this work, we discussed integrable aspects of critical topologically massive gravity in three dimensions. We first showed that the partition function of the logarithmic sector of the theory is a generating function of the Burgers hierarchy in its potential form. The Burgers hierarchy is a family of nonlinear evolution equations possessing infinitely many symmetries or flows, all of which preserve the first member of the family, the Burgers equation. The latter was originally used to describe the mathematical modelling of turbulence [21], a phenomenon also described in LCFT related works [89–93]. Furthermore, Burgers equation has shown relevance in identifying a relationship between (Burgers) turbulence and disordered systems via a mapping between Burgers equation and the problem of a directed polymer in a random medium [94]. Systems with quenched disorder have been described by the type of $c = 0$ LCFT that arises naturally in critical TMG [95–97]. The results obtained in our work related to Burgers hierarchy could therefore be of interest for potential $AdS_3/LCFT_2$ applications.

We then showed that the partition function of the logarithmic sector is a generating function of the integrable hierarchy of Kadomtsev-Petviashvili (KP) nonlinear partial differential equations, also called a KP $\tau$-function. The KP hierarchy was originally studied as equations with soliton solutions, and the fact that the partition function is a KP soliton $\tau$-function brings to light the solitonic aspect of the logarithmic fields counted in $Z_{log}$.

The important role played by $\tau$-functions of integrable systems in the moduli theory of Riemann surfaces was illustrated in our work, through the link between integrability and Hurwitz numbers. In this paper, as a $\tau$-function of the KP hierarchy, we showed that $Z_{log}(q, \bar{q})$ is also a generating function of Hurwitz numbers. As such, it can be understood as a function enumerating genus zero covering spaces. A description of the target space as a flat branched

Riemann surface of revolution was given, based on the fact that the logarithmic field $\psi_{\mu\nu}^{new}$ is composed of a complex logarithm function, and can be interpreted as a branch point field associated to the branch point $\mu l = 1$. Such fields with branch cut singularities on Riemann surfaces have been encountered in many applications of conformal field theory, and have been dealt with by utilizing a suitable covering of the Riemann surface on which the fields become singlevalued. Such a method was actually used in the earliest work on logarithmic conformal field theory [30].

Hurwitz partition functions have also appeared in the literature as partition functions for HOMFLY polynomials of some knot in the theory of Ooguri-Vafa (OV) [98–100]. In particular, for any torus knot, the OV partition function is a $\tau$-function of the KP hierarchy [101]. This shows a deep connection between Hurwitz $\tau$-functions and 3d Chern-Simons theory, and it is agreeable to see this subtle connection also arises in our work.

The solitonic nature of the logarithmic fields coupled with the fact that they are associated with conical singularities of excess angle (multiples of $2\pi$) on a Riemann surface confirms the expectation of conical spaces such as the ones studied in [102] to play a role in this Chern-Simons formulation of a nonunitary three-dimensional gravity.

To the best of our knowledge, our results have not appeared in the literature in the context of topologically massive gravity at the critical point, and are therefore novel. Several points however, still remain unclear. For instance, an important issue yet to be clarified is the 1-loop exactness of the partition function in critical TMG. This point, which would shed some light on the understanding of the instanton contributions to the full partition function, has not been addressed in our work. We hope to discuss it in future.

## Acknowledgements

The author would like to thank Alexei Morozov for correspondence on Hurwitz numbers and one-part Schur polynomials. It is also a pleasure to acknowledge the referees, whose constructive comments helped in improving the quality of the paper. The author was supported by the University of Pretoria's Postdoctoral Fellowship Programme.

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
