# Peer review of "Integrable hierarchies, Hurwitz numbers and a branch point field in critical topologically massive gravity"

_SciPost Physics Core, doi:SciPost Phys. 12, 132 (2022)_

## Round 1 · Referee Report · Daniel Grumiller (Referee 1) · 2021-9-28

Strengths

  1. The main result that the partition function associated with log excitations in cTMG is a KP tau-function is clear, succinct, yet unexpected (at least to me) and novel.

  2. Relatedly, also the interpretation of this partition function as generating function for Hurwitz numbers is both surprising and intriguing.

  3. The introduction does an excellent job in reviewing essential aspects of AdS/log CFT in the context of cTMG.

Weaknesses

  1. Some pertinent questions remain unanswered. For instance, it is unclear whether or not the partition function in cTMG is 1-loop exact. Perhaps this is the case, and if so, the structures uncovered in this paper may be a first indication for 1-loop exactness. However, this is far from clear.

  2. Similarly, it would be nice to also understand the instanton contributions to the full partition function. Is it possible to simply take a sum over modular images, like in Ref. [6], or not? As the paper makes not statement about 1-loop exactness or instanton contributions, this is not a criticism of the way the paper is written, but more an encouragement to eventually take the next step.

  3. Some of the jargon was unfamiliar to me. While the paper gave all the required pointers to the literature and explained all expressions that were new to me, it is still somewhat difficult to read.

Report

The paper meets all the acceptance criteria.

It addresses an important problem in cTMG, namely to get better insight into the 1-loop partition function, with potential applications for AdS/log CFT. The main research result, that this partition function is the KP tau-function, is a significant and unexpected advance in this research field.

My only criticism is that some of the jargon was unfamiliar to me (and by extrapolation could be unfamiliar to many readers), so we might have benefited from additional explanations or simple examples - for instance, the discussion in section 3 on Hurwitz numbers requires a couple of additional definitions (like "ramification" or "branched covering"), and they just follow one after the other without providing more context. While the clarity of the paper is still good, I think with a bit of additional effort it could be made even clearer.

Requested changes

  1. Go as easy on the jargon as possible and try to find simple examples or provide additional explanations for less familiar terms. To give just one example, a sentence like "By definition, the Plücker embedding of the Grassmannian is given by quadratic equations called Hirota bilinear equations." is only understandable if it is clear to the reader what a Plücker embedding is. Granted, this is briefly defined before, but for a reader encountering this term for the first time it could be helpful to give a bit more context. There are similar examples throughout the text.

  2. Optionally, perhaps add some outlook in section 5 pointing to some of the unresolved issues concerning the partition function (see points 1. and 2. under "Weaknesses")

---

## Round 1 · Referee Report · Luca Ciambelli (Referee 2) · 2021-12-30

Strengths

1- well written and detailed

2- Important review section of mathematical tools

3- Interesting new results on the logarithmic contribution to the partition function of critical TMG

Weaknesses

In my view, the main weakness is that the paper contains almost 8 pages of (very well-written) review of mathematical background material, whereas the main result is presented in a quick and dismissal way in the last 2 pages. I suggest to keep the first review part as it is, but to spend considerably more time explaining the original result on the logarithmic contribution to the partition function. What is the state of art on this topic? Why the author's new original result is important? While this paper is supposed to be about critical TMG, the main body of the paper is on an a priori different topic, so I believe it is important to dedicate more time to the ''TMG side''.

The author uses a lot of technical expressions and words, which sometimes make the manuscript not accessible and hard to follow. The paper is dense, with a lot of technical concepts dropped (see the paragraph below eq. (18), as an indicative example). The author should clarify, expand more, and make this work more accessible to the reader.

Report

I believe this paper is well written and meets all the criteria to be published in this journal, once the minor revisions requested below are addressed. Therefore, I am happy to review the revised version of this paper.

Requested changes

1- Dilute the paper and spend more time on the new original results.

2- Beginning of page 2: the comment on ''deforming pure Einstein gravity'' is confusing. First of all, Einstein gravity with negative cosmological constant has still no propagating degrees of freedom in 3d. Second, the presence of black holes is not relating to propagating degrees of freedom, and indeed also Einstein gravity with vanishing cosmological constant has black holes solutions (Banados solutions). I would refrain from putting on equal footing Einstein gravity with negative cosmological constant and TMG.

3- The terminology used by the author is often not the standard one (e.g. ''cosmological TMG'', or ''Fefferman-Graham-Brown-Henneaux'', or ''Lie Heisenberg algebra''). In a such a dense paper (see weaknesses), this makes it harder to read.

4- Please define q and qbar in eq. (2).

5- Please define in the Introduction Hurwitz numbers and Burgers hierarchies, since they are the core of the paper

6- Please add references in the first paragraph of section 2 and the first paragraph of section 2.1.

7- Please define notation in equation (14)

8- Please explain more below (16)

9- Please expand the first paragraph in section 2.2.2

10- Please explain better how the assignment (25) defines the embedding (26)

11- Unclear notation in (29) and (30) for the ''z''. In the previous equations, ''z'' had a superscript, while in these equations it has a subscript.

12- The symbol for the bosonic Fock space is undefined

13- Please expand the paragraph below eq. (42)

14- Since, as already pointed out, the manuscript is dense, I suggest to write a recap at the end of section 4 of the main equations and main conclusions.

15- Please provide at the end of the conclusions some possible outlook of this work, to put it in context and localize it in the vast literature on the subject. This is also important to show the author's far-reaching research lines and goals.

---

## Round 2 · Referee Report · Luca Ciambelli (Referee 2) · 2022-2-1

Report

I thank the author for scrupulously addressing all the issues I mentioned in the first report. I believe the amended version meets the publication criteria. Nonetheless, I would like to point out just an aesthetic-related problem concerning references. I suggest that the author corrects certain citations, e.g. citation [8], to uniformize the style and make it easier for the reader to identify the cited papers.

---

## Round 2 · Referee Report · Daniel Grumiller (Referee 1) · 2022-2-25

Report

While the strengths remain the same as in my first report, the weaknesses were addressed in the amended manuscript. Therefore, I suggest publication of the paper in its present form.

---

## Round 2 · Author Response

We thank both referees for their careful reading of the first version of our manuscript, and for their constructive comments, all of which were addressed in the second version of the manuscript. Below, we review the changes made according to the requests in the two reports.

Report 1:

  1. Go as easy on the jargon as possible and try to find simple examples or provide additional explanations for less familiar terms. To give just one example, a sentence like "By definition, the Plücker embedding of the Grassmannian is given by quadratic equations called Hirota bilinear equations." is only understandable if it is clear to the reader what a Plücker embedding is. Granted, this is briefly defined before, but for a reader encountering this term for the first time it could be helpful to give a bit more context. There are similar examples throughout the text.

Some definitions were provided with examples and graphical illustrations in the amended manuscript. An example of the meaning of Plücker embedding was given under Eq. (26) in the first version, with a paragraph between Eq. (26) and Eq. (32) in the amended version. Two figures were drawn below the definitions of covering and branched covering, giving a graphical illustration to those definitions. A paragraph was added to explain a bit more in detail how the connected and disconnected Hurwitz numbers appear in the plethystic logarithm and plethystic exponential, respectively. As such, the paragraph between Eq. (41) and Eq. (43) in the first version has been replaced by a longer paragraph between Eq. (47) and Eq. (53) in the amended version.

  1. Optionally, perhaps add some outlook in section 5 pointing to some of the unresolved issues concerning the partition function (see points 1. and 2. under "Weaknesses")

Some outlook was added in the very last paragraph of section 5 pointing to the issues that still need clarification, such as the 1-loop exactness of the partition function and the instanton contribution to the full partition.

Report 2:

1- Dilute the paper and spend more time on the new original results.

The amended version is an extended version of the first one, with an attempt to make the latter less concentrated.

2- Beginning of page 2: the comment on ''deforming pure Einstein gravity'' is confusing. First of all, Einstein gravity with negative cosmological constant has still no propagating degrees of freedom in 3d. Second, the presence of black holes is not relating to propagating degrees of freedom, and indeed also Einstein gravity with vanishing cosmological constant has black holes solutions (Banados solutions). I would refrain from putting on equal footing Einstein gravity with negative cosmological constant and TMG.

The paragraph at the beginning of page 2 in the first version was modified in the new version of the manuscript. We hope that the amended paragraph at the same location in the new version will clear potential confusions concerning Einstein gravity with negative cosmological constant and TMG being put on equal footing.

3- The terminology used by the author is often not the standard one (e.g. "cosmological TMG", or "Fefferman-Graham-Brown-Henneaux", or "Lie Heisenberg algebra"). In a such a dense paper (see weaknesses), this makes it harder to read.

The term "Fefferman-Graham-Brown-Henneaux" was inspired by Ref. [9], in which it appears in the introduction. We have added Ref. [9] next to the term in the amended version. We respectfully acknowledge the referee's comment and agree that such terminology has not appeared much in the literature. In connection with the term "cosmological TMG", we attempted, perhaps without much precision, to remain faithful to the terms encountered in the first papers that discuss log. gravity, such as Ref. [10] in the amended version of our manuscript. In that spirit, and with respect to the referee's comment, we hope the referee will allow us to proceed with the use of that specific term.

4- Please define q and qbar in eq. (2).

q and qbar were defined in Eq. (2) of the amended version of the manuscript.

5- Please define in the Introduction Hurwitz numbers and Burgers hierarchies, since they are the core of the paper

A definition of Hurwitz numbers was added in the introduction. A definition of Burgers hierarchy was also added, with Ref. [20-23] included. The introduction was also expanded with the definition the KP hierarchy and added historical and recent references (Ref. [24-26]) which provide a way to contextualize our work in the literature.

6- Please add references in the first paragraph of section 2 and the first paragraph of section 2.1.

Ref. [34-43] were added in the the first paragraph of section 2, and Ref. [44-46] in the first paragraph of section 2.1.

7- Please define notation in equation (14)

Notation was defined in Eq. (14) of the amended version of the manuscript.

8- Please explain more below (16)

Some explanation was provided below Eq. (16) in the amended version of the manuscript.

9- Please expand the first paragraph in section 2.2.2

The first paragraph in section 2.2.2 of the amended version of the manuscript was expanded.

10- Please explain better how the assignment (25) defines the embedding (26)

In order to explain how the assignment in Eq. (25) defines the embedding of Eq. (26), an example was provided below Eq. (26), up to the sentence below Eq. (32).

11- Unclear notation in (29) and (30) for the "z". In the previous equations, "z" had a superscript, while in these equations it has a subscript.

We thank the referee for allowing us to spot this typo in Eqs. (29) and (30) of the first manuscript. The typo was fixed in Eqs. (35) and (36) of the amended version of the manuscript.

12- The symbol for the bosonic Fock space is undefined

The symbol for the bosonic Fock space was defined in the first sentence of the second paragraph under "Boson-fermion fermion correspondence", page 9 of the amended manuscript.

13- Please expand the paragraph below eq. (42)

The paragraph below Eq. (42) in the first version of the manuscript was modified to explain a bit more in detail how the connected and disconnected Hurwitz numbers appear in the plethystic logarithm and plethystic exponential, respectively. As such, the paragraph between Eq. (41) and Eq. (43) in the first version has been replaced by a longer paragraph between Eq. (47) and Eq. (53) in the amended version.

14- Since, as already pointed out, the manuscript is dense, I suggest to write a recap at the end of section 4 of the main equations and main conclusions.

A recap of our results was given at the end of section 4, pointing to important equations and the connection between the results.

15- Please provide at the end of the conclusions some possible outlook of this work, to put it in context and localize it in the vast literature on the subject. This is also important to show the author's far-reaching research lines and goals.

Section 5 was renamed and extended. Potential applications for nonunitary holography are mentioned from connections between Burgers equations and disordered systems. The importance of the interpretation of the log. partition function as a Hurwitz tau function, which opens exploration in new directions is also discussed. Despite some progress made in understanding the log. partiton function, the limitation in our work over the past years is also mentioned, with the example of the 1-loop exactness of the partition function not yet clarified.

---

## Editorial Decision

published